DOI: 10.1038/s41467-017-00111-8　　**OPEN**

# Gelsolin dysfunction causes photoreceptor loss in induced pluripotent cell and animal retinitis pigmentosa models

Roly Megaw[1], Hashem Abu-Arafeh[1], Melissa Jungnickel[2], Carla Mellough [3], Christine Gurniak[4], Walter Witke[4], Wei Zhang[5], Hemant Khanna[5], Pleasantine Mill[2], Baljean Dhillon [6], Alan F. Wright[2], Majlinda Lako[3] & Charles ffrench-Constant[1]

Mutations in the Retinitis Pigmentosa GTPase Regulator (*RPGR*) cause X-linked RP (XLRP), an untreatable, inherited retinal dystrophy that leads to premature blindness. RPGR localises to the photoreceptor connecting cilium where its function remains unknown. Here we show, using murine and human induced pluripotent stem cell models, that RPGR interacts with and activates the actin-severing protein gelsolin, and that gelsolin regulates actin disassembly in the connecting cilium, thus facilitating rhodopsin transport to photoreceptor outer segments. Disease-causing *RPGR* mutations perturb this RPGR-gelsolin interaction, compromising gelsolin activation. Both *RPGR* and *Gelsolin* knockout mice show abnormalities of actin polymerisation and mislocalisation of rhodopsin in photoreceptors. These findings reveal a clinically-significant role for RPGR in the activation of gelsolin, without which abnormalities in actin polymerisation in the photoreceptor connecting cilia cause rhodopsin mislocalisation and eventual retinal degeneration in XLRP.

[1] MRC Centre for Regenerative Medicine, University of Edinburgh, 5 Little France Drive, Edinburgh EH16 4UU, UK. [2] MRC Human Genetics Unit, Institute for Genetics and Molecular Medicine, University of Edinburgh, Crewe Road, Edinburgh EH4 2XU, UK. [3] Institute of Genetic Medicine, Newcastle University, Newcastle NE1 3BZ, UK. [4] Institute fur Genetik, Universitat Bonn, Karlrobert-Kreiten-Strasse. 13, 53115 Bonn, Germany. [5] Department of Ophthalmology, UMASS Medical School, 368 Plantation St, Albert Sherman Center, AS6-2043, Worcester, Massachusetts 01605, USA. [6] Centre for Clinical Brain Sciences, University of Edinburgh, Chancellor's Building, 49 Little France Crescent, Edinburgh EH16 4SB, UK. Correspondence and requests for materials should be addressed to R.M. (email: roly.megaw@ed.ac.uk)

The rod photoreceptor enhances processing of visual stimuli by compartmentalising proteins critical for phototransduction within its outer segment (OS). The OS emerges from the distal end of the connecting cilium (CC), with membrane extensions folding to form thousands of disc-like processes that stack to form the body of the OS. The CC is therefore a highly specialised primary cilium whose protein composition is unique to the retina. Up to 10% of OS discs are renewed every day[1] and, with all photoreceptor proteins being synthesised in the cell's inner segment (IS), this rate of OS turnover requires high levels of protein trafficking from the IS to (and across) the CC to maintain homeostasis. Indeed, up to 1000 molecules of rhodopsin are trafficked through the 0.3 μm-wide CC in human photoreceptors per second[2]. Breakdown of this cilia trafficking results in protein mislocalisation and, eventually, photoreceptor death. Such photoreceptor degeneration is the hallmark of retinitis pigmentosa (RP)[3], a heterogenous group of inherited retinal dystrophies affecting 1 in 3000 people[4]. RP causes severe visual loss and blindness. *RPGR* mutations account for 70–90% of XLRP and 10–15% of all RP[4]. Whilst its exact function remains unknown, RPGR localises to the base of the CC where it was proposed to play a role in trafficking of rhodopsin to the OS based on knockout mouse models[5]. In addition, depletion of *RPGR* in cell lines and mice increases actin polymerisation[6, 7, 8]. To define RPGR's role in photoreceptor maintenance and to investigate the molecular pathogenesis of XLRP, we generated induced pluripotent stem cells (iPSCs) from patients with *RPGR* type XLRP (*RPGR*/XLRP).

Here, we show *RPGR*-mutant, iPSC-derived photoreceptor cultures display increased actin polymerisation, a phenotype also observed in the RPGR knockout mice that show rhodopsin mislocalisation and photoreceptor degeneration. RPGR binds to and activates the actin-severing protein gelsolin, an interaction that is lost in RPGR-mutant (XLRP) cells, and *Gelsolin* knockout mice also show abnormalities of actin polymerisation and rhodopsin localisation. Activated Gelsolin rescues the ciliogenic defect observed in RPGR-depleted cells. We conclude, therefore, that *RPGR* mutations in XLRP lead to defective gelsolin activation and defects in actin regulation and rhodopsin trafficking within the photoreceptor CC.

## Results

**iPSCs as a novel model for RPGR/XLRP.** Induced pluripotent stem cells (iPSCs) were generated from fibroblasts of two brothers carrying an *RPGR* mutation (g.ORF15+689–692del4) using previously described methods[9, 10] (Supplementary Fig. 1a, b). Control iPSCs were generated from an unaffected son/nephew. All iPSC clones expressed markers of pluripotency, as demonstrated with qPCR (Supplementary Fig. 1c) and immunostaining (Supplementary Fig. 1d). iPSC lines could differentiate into all three major cell lineages; ectoderm, mesoderm and endoderm (Supplementary Fig. 1e). To generate photoreceptors from these iPSC clones, a three-dimensional retinal differentiation protocol was optimised[11, 12] (Fig. 1 and Supplementary Fig. 2). Successful patterning of free floating embryoid bodies resulted in spontaneous outpouching of optic vesicles (Fig. 1a, b), which invaginated into optic cup-like structures (Fig. 1c), as previously described[11]. After 100 days, retinal pigment epithelium (RPE) emerged on the outer surface of the cups (Fig. 1d) while inside an organised layer of photoreceptors emerged, expressing several maturity-associated markers (Fig. 1e, f). These cells displayed cilia with the classic "9 + 0" microtubule doublet organisation characteristic of primary cilia (Fig. 1g, h) and also were associated with additional membranous, disc-like material produced and deposited between the melanosome-containing RPE cells (Fig. 1j). RPGR, as previously reported[5], localised to this emerging 'CC' that joins the cell body to the rod OS (Fig. 1i). We concluded that human photoreceptors, derived from iPSCs, could therefore be used to study the molecular pathogenesis of *RPGR*/XLRP.

**RPGR regulates actin polymerisation in hiPS derived and rodent photoreceptors.** We first asked whether the abnormalities in the actin cytoskeleton reported in *RPGR*-depleted cell lines[6] are present in iPSC-derived photoreceptors from *RPGR*/XLRP subjects. Increased actin polymerisation was observed in cultured iPSC-derived photoreceptors of both patients compared to the control (Fig. 2a–d, Supplementary Fig. 3a). The iPSC-derived *RPGR*/XLRP photoreceptor cultures (CB10 and HB02) showed a 2.7-fold (SEM 0.127; $p < 0.0001$, $n = 3$) and 2.49-fold (SEM 0.34; $p = 0.0081$, $n = 3$) increase in actin polymerisation, respectively,

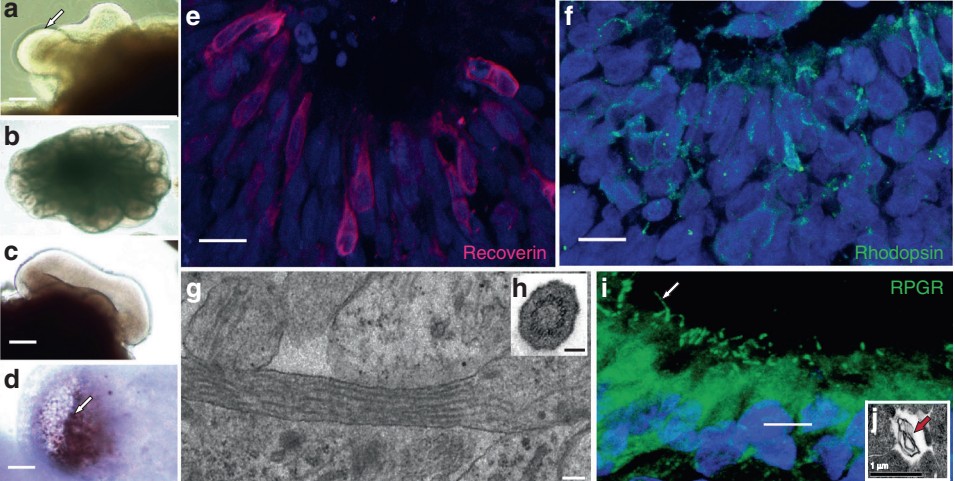

**Fig. 1** Three-dimensional patterning of induced pluripotent stem cells produces mature photoreceptors. Free floating aggregates grown using a retinal differentiation protocol[10, 11] self-organise into budding spheroids with a single bud shown in **a** (*arrow*) and multiple buds from one spheroid in **b**. Buds invaginate (**c**) to form optic cups which mature over 100 days (**d**). Pigmented retinal pigment epithelium (RPE) emerges externally (*arrow*; **d**) while a radial arrangement of Recoverin (**e**) and Rhodopsin (**f**) expressing photoreceptors organise internally, forming an outer nuclear-like layer. Electron microscopy studies reveal a classic "9 + 0" microtubule doublet formation in these cilia (**g**, inset **h** shows cross-section). RPGR is present in photoreceptor cilia (*arrow*; **i**) whilst electron microcopy shows the production of membranous material, as required for outer segment formation (*arrow*, **j**). *Scale bars*: 400 μm (**b**); 200 μm (**a**); 50 μm (**c**, **d**); 10 μm (**e**, **f**); 100 nm (**g**); 200 nm (**h**); 5 μm (**i**); 1 μm (**j**)

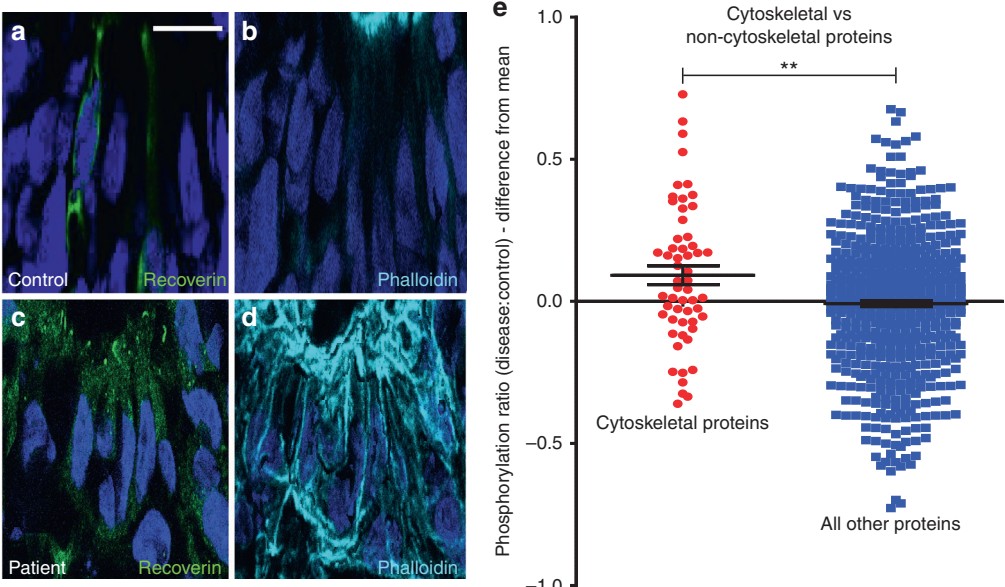

**Fig. 2** *RPGR*-mutant, iPSC-derived, 3-dimensional photoreceptor cultures display perturbed actin regulation. **a–d** *RPGR*-mutant photoreceptors display increased actin polymerisation, as evidenced by increased phalloidin binding in the Recoverin-positive photoreceptors of patient-derived cultures (panels **c**, **d**) as compared to photoreceptors from the control patient (panels **a**, **b**). See text for details of quantification of pixel intensities. Nuclei stained with Hoechst (*blue*). **e** The mean phosphorylation level of cytoskeletal regulatory proteins was higher in *RPGR*/XLRP photoreceptors compared to control photoreceptors (*red dots*; $0.092 \pm 0.0033$, $n = 56$ proteins). This was significantly more than the mean ratio of phosphorylation levels of non-cytoskeletal proteins when *RPGR*/XLRP photoreceptors were compared to control (*blue dots*; $-0.008 \pm 0.01$, $n = 583$ proteins, unpaired, two-tailed *t*-test, $p = 0.004$ (** = $p < 0.01$))

compared with the control line (MB02), as determined by quantitative imaging of phalloidin staining (unpaired, two-tailed *t*-test). As an alternative method of confirming a cytoskeletal abnormality, iPSC-derived photoreceptor cultures were screened for phosphorylation levels of 639 phosphoproteins (Fig. 2e). Unbiased proteomic analysis of *RPGR*/XLRP and control lines revealed a significant increase in phosphorylation of cytoskeletal regulatory proteins in the diseased photoreceptor cultures as compared to control. The mean ratio of phosphorylation level in diseased vs control cells was $0.092 \pm 0.0033$ ($n = 56$ proteins) compared with non-cytoskeletal proteins ($-0.008 \pm 0.01$, $n = 583$ proteins, $p = 0.004$) (Fig. 2e, Supplementary Data 1). These results further support a disruption of cell signalling pathways regulating actin turnover in *RPGR*/XLRP mutant photoreceptors.

In parallel experiments to confirm a role for RPGR in actin turnover in the photoreceptor CC in vivo, we examined a recently reported *Rpgr* knockout (KO) mouse[13]. Significant photoreceptor degeneration was observed by 4 months of age (Fig. 3a–c). Prior to this, at 3 weeks of age, signs of photoreceptor stress were apparent as evidenced by increased GFAP immunolabeling throughout the radial length of Müller cells in the outer nuclear layer (ONL: Fig. 3d). In addition, the photoreceptors failed to correctly localise rhodopsin to the OS at this stage: instead it was trafficked to the outer plexiform layer and eventually remained in the peri-nuclear space (Fig. 3e, f). Cytoskeletal examination in murine retina during development (at postnatal day 2 and 10) revealed no difference between wild type and *Rpgr* KO photoreceptors (Supplementary Fig. 4), suggesting the initial photoreceptor development was RPGR-independent at stages where neonatal eyes are still closed and phototoxicity minimal. The degeneration in *Rpgr*-mutant photoreceptors seems to be linked to when eyes open at P14, as reactive gliosis is detected by P21 and by 2 months of age increased actin polymerisation was seen in the connecting cilia (Fig. 3g), all prior to photoreceptor loss. These results demonstrate that RPGR is required for actin regulation in the mature photoreceptor CC and for correct localisation of rhodopsin to its OS in vivo.

**RPGR mutations ameliorate the activation of the actin-severing protein Gelsolin.** Next, we performed a cytoskeletal-focused phosphoarray in an attempt to determine the mechanism by which RPGR regulates actin turnover. We observed in *RPGR*/XLRP photoreceptor cultures that cofilin, a protein whose role in regulating actin polymerisation appears important for ciliogenesis[14] and which localises to the photoreceptors in mature wild-type retina (Supplementary Fig. 5), showed 3.16-fold higher levels of phosphorylation of Serine 3, a post-translational modification that inhibits binding to and treadmilling/depolymerisation of F-actin[15] (Fig. 4a). This result was confirmed by western blot analysis of repeat cultures of both *RPGR*/XLRP lines compared to control (Fig. 4b; blot density measurement control MB02 22.544% vs patient CB10 40.364% vs patient HB02 37.092%). This post-translational modification of cofilin is normally inhibited by the actin-severing protein gelsolin[16], suggesting that loss of gelsolin activity might be responsible for the increased actin polymerisation and photoreceptor abnormalities seen in the *RPGR*-mutant photoreceptors. In support of this, it has been shown that fibroblasts from *gelsolin* knockout mice show similar increased actin fibre formation[17] as seen in our cultured iPSC-derived *RPGR*/XLRP photoreceptors and *RPGR*-knockdown lines[6]. Moreover, gelsolin has been shown to regulate cilia formation, as evidenced by degeneration of outer ear stereocilia in *gelsolin* KO mice[18] and the identification of gelsolin in an siRNA screen for genes required in ciliogenesis[19]. We therefore examined gelsolin activation in human *RPGR*/XLRP photoreceptors. Gelsolin occurs in two forms; a closed, inactive conformation in which the amino- and carboxy-terminal halves bind to each other, and an open, active conformation in which the N-terminal half binds to and cleaves actin filaments[20, 21]. The two forms can be separated by biochemical isolation of F-actin, to which only the activated form will bind[22, 23]. When F-actin was isolated from our cultures, it revealed decreased levels of bound (active) gelsolin in *RPGR*/XLRP iPSC-derived photoreceptors from both XLRP subjects compared to control photoreceptors, despite increased levels of the polymerised F-actin substrate

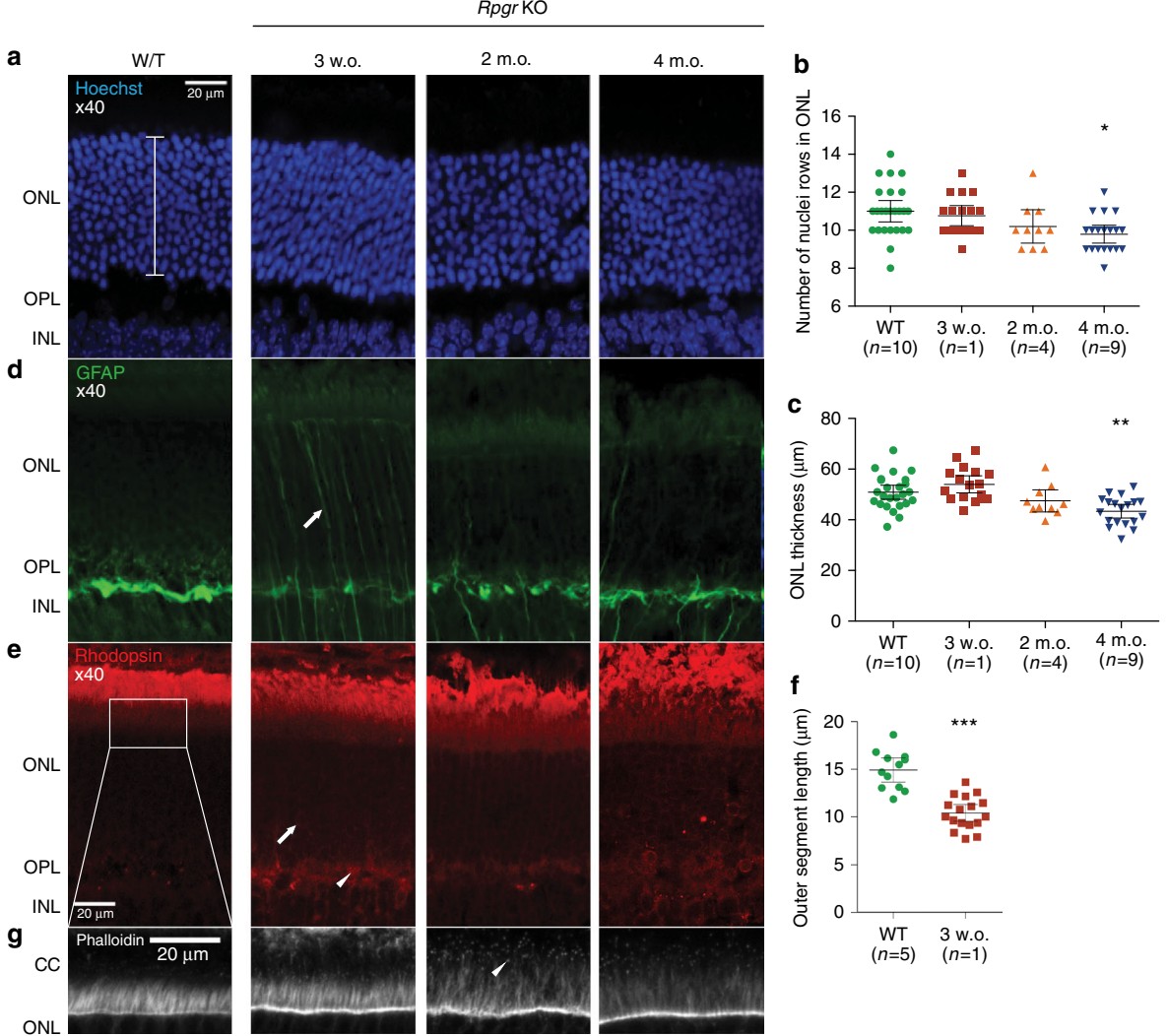

**Fig. 3** The *Rpgr* KO mouse retina demonstrates actin dysregulation and rhodopsin mislocalisation in the photoreceptor layer prior to degeneration.
**a–c** Outer nuclear layer (ONL—photoreceptor) degeneration develops in the *Rpgr* KO mouse (*bracket* represents the region measured). This change is significant by 4 months of age (compared to 4-month old wild-type) as indicated by the number of rows of photoreceptor nuclei (**b**; Figures denote mean ±SEM, Kruskal–Wallis test [$H = 30.8$, 4 d.f., $p < 0.0001$], Dunn's multiple comparisons test, $p < 0.05$, * $= p < 0.05$) and ONL thickness (**c**; Figures denote mean±SEM, one-way ANOVA [$F(4,76) = 10.7$, $p < 0.0001$], Tukey's multiple comparisons test, $p < 0.01$ ** $= p < 0.01$). The WT control shown is 4 months of age. **d**, **e** Reactive gliosis as reflected by increased GFAP immunolabeling throughout the radial length of Müller cells in the ONL is apparent as early as 3 weeks of age (arrow; **d**) while rhodopsin, normally only present in the outer segments outside (above in the figure) the ONL, is now mislocalised into other regions of the photoreceptor in the outer plexiform layer (OPL—*arrowhead*; **e**) and peri-nuclear area (*arrow*; **e**) at this timepoint, prior to the visible onset of photoreceptor degeneration. **f** Outer segment length is reduced by 3 weeks of age (compared to 4-month old wild-type; Figures denote mean±SEM, *** $= p < 0.001$) after which they become markedly disorganised. **g** Increased F-actin is seen in the connecting cilium (*arrowhead*) of the *Rpgr* KO mouse photoreceptor compared to wild type (Images representative of $n = 1$–10 analysed)

(Fig. 4c; blot density measurement control MB02 15.507% vs patient CB10 7.316% vs patient HB02 8.012%, Supplementary Fig. 3b). These observations show that *RPGR*/XLRP mutations attenuate gelsolin activation with subsequent disruption of its downstream pathways.

***Gelsolin* KO mice phenocopy the retinal abnormalities seen in *Rpgr* KO mice.** To confirm that the increased actin polymerisation and rhodopsin mislocalisation observed in *Rpgr* KO photoreceptors could be the consequence of loss of gelsolin activity, we examined the retina of the *gelsolin* KO mouse[17]. At 5 months of age, a similar phenotype was seen, with rhodopsin mislocalisation to the outer plexiform layer and peri-nuclear space (Fig. 5a).

Increased GFAP immunolabeling throughout the radial length of Müller cells in the ONL was observed (Fig. 5d). Further, degeneration of the photoreceptor-containing ONL was significant by 5 months of age (Fig. 5b, c). At earlier time points (2-month old), increased actin polymerisation was seen in the connecting cilia (Fig. 5e), akin to the *Rpgr* KO mouse. Together, these observations show that gelsolin regulates actin polymerisation in the photoreceptors and is required for both rhodopsin transport to the photoreceptor OS and photoreceptor maintenance.

**RPGR binds gelsolin and activated gelsolin can rescue RPGR-deficient cells.** Having established that loss of gelsolin phenocopies key abnormalities of the *Rpgr* KO mouse, we next sought

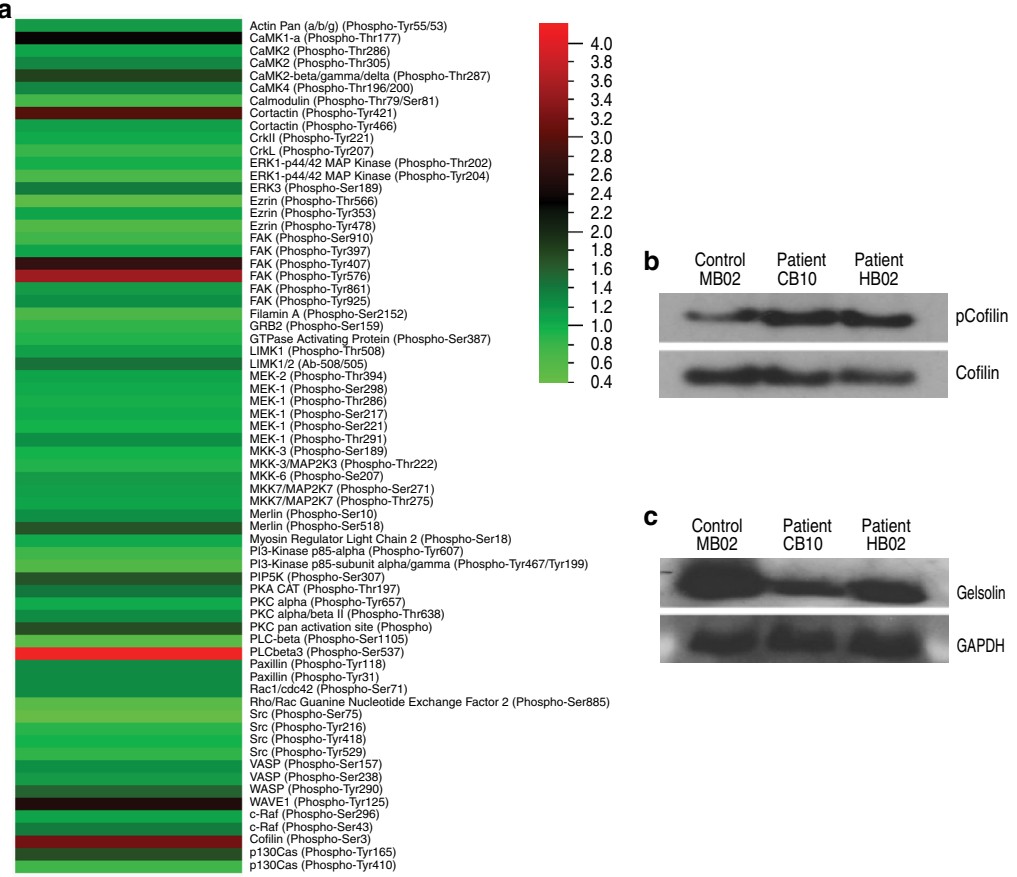

**Fig. 4** Studies of iPSC-derived photoreceptor cultures reveal perturbation of gelsolin and cofilin activity in *RPGR*-mutant lines. **a** A phosphoarray comparing phosphorylation levels of cytoskeletal proteins in patient and control photoreceptor cultures identified cofilin as hyperphosphorylated on Serine 3 in *RPGR*-mutant photoreceptors (results presented as a heat map, with a patient:control ratio of phosphorylation for cofilin of 3.16). Note that only 3 other proteins (cortactin, FAK and PLCbeta3) show similar levels of hyperphosphorylation. **b** This increase in cofilin phosphorylation was confirmed by western blotting of repeat cultures using a phospho-serine 3 specific antibody. **c** F-actin-bound (active) gelsolin is reduced in *RPGR*-mutant photoreceptors

to determine whether there is a direct biochemical interaction between RPGR and gelsolin. Co-immunoprecipitation (Co-IP) of recombinant proteins revealed that gelsolin binds directly to the ubiquitous splice variant (RPGR$^{Ex1-19}$) but not the retina-enriched RPGR$^{ORF15}$ form of the protein (Supplementary Fig. 6). To confirm the RPGR-gelsolin interaction in vivo, co-IP and reverse co-IP using bovine retinal extract were performed. These experiments revealed such an interaction between the two proteins in vivo (Fig. 6a), as did endogenous co-IP of protein product from photoreceptor cultures derived from our control iPSC lines (Fig. 6b). In these endogenous retinal extracts, however, both the ubiquitous splice variant (RPGR$^{Ex1-19}$) and the retina-enriched RPGR$^{ORF15}$ isoforms were pulled down with gelsolin. Importantly, this interaction was disrupted in photo-receptor cultures derived from both our *RPGR*/XLRP iPSC lines (Fig. 6b), with both the ubiquitous splice variant (RPGR$^{Ex1-19}$) and the retina-specific RPGR$^{ORF15}$ forms of the protein no longer binding to gelsolin in the presence of the mutation in the latter. Thus RP-causing mutations in the retinal-specific RPGR$^{ORF15}$ block the interaction of the ubiquitous RPGR$^{Ex1-19}$ isoform with gelsolin, resulting in reduced gelsolin activation and impaired F-actin turnover.

If the RPGR-gelsolin interaction were responsible for regulating gelsolin activity, and thus actin polymerisation in the photoreceptor, we would predict that the consequences of *Rpgr* loss could be rescued by expression of activated gelsolin. To test this, we examined an easily-quantifiable reporter of RPGR function; the inhibition of ciliogenesis induced by serum starvation in *RPGR*-depleted hTERT-RPE cells (Fig. 6c–e) as previously described[6]. Concurrent expression of the active, N-terminal half of gelsolin[24] in this cell line restored cilia formation to wild-type levels (Fig. 6d–e), confirming that gelsolin activation can rescue *RPGR* loss.

## Discussion

This report shows that RPGR functions in photoreceptors to activate the actin-cleaving protein gelsolin, since KO mice lacking gelsolin showed the same retinal phenotype as *RPGR* KO mice and human XLRP mutations in *RPGR* prevented the normal interaction between RPGR and gelsolin, reducing its activation and increasing actin polymerisation. Further support for the significance of this interaction comes from a report that families with a homozygous D187N (G654A) mutation in the human gelsolin (*GSN*) gene have RP, as well as manifestations of systemic amyloidosis[25] resulting from either loss-of-function (e.g. RP) or deposition of amyloidogenic gelsolin fragments.

This activation of gelsolin is achieved by direct binding to RPGR, as evidenced by Co-IP studies showing the interaction of gelsolin with RPGR and by disruption of both this binding and subsequent activation of gelsolin in the presence of an XLRP mutation. Our results suggest the primary binding site appears to be in the ubiquitous splice variant (RPGR$^{Ex1-19}$) rather than the retina-specific RPGR$^{ORF15}$ isoform. The finding that the pathogenic mutation that prevents this binding is in the RPGR$^{ORF15}$

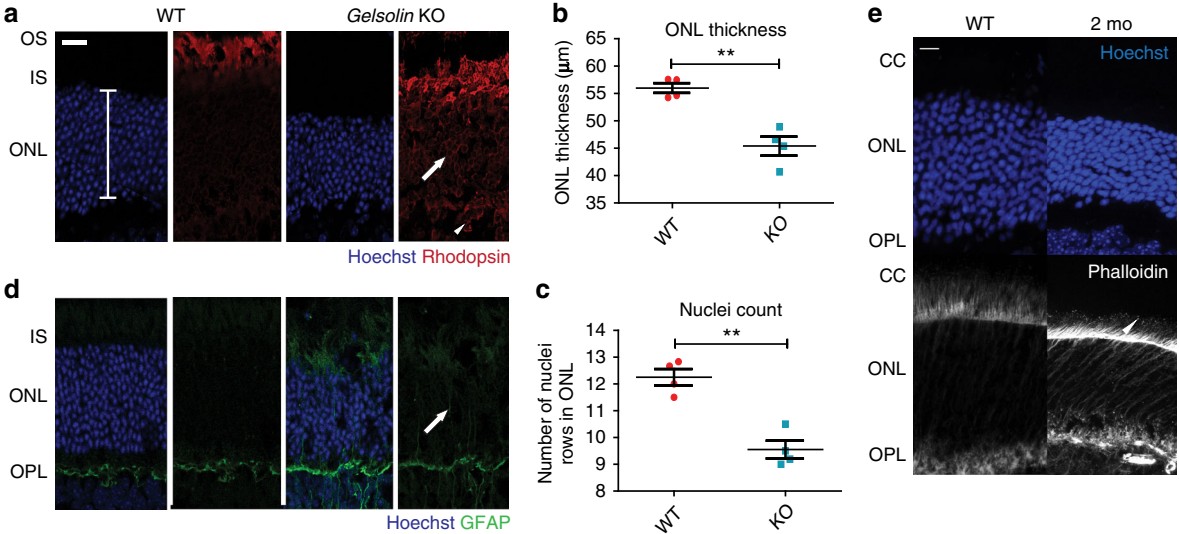

**Fig. 5** Knockout mouse studies confirm a role for Gelsolin in photoreceptor maintenance. **a**–**c** Outer nuclear layer (ONL) degeneration develops in the *gelsolin* knockout mouse at 5 months of age (**a**; *bracket* represents ONL length measured); significant for ONL thickness (**b**; $n = 4$, Figures denote mean ±SEM, unpaired two-tailed $t$-test, ** = $p < 0.01$) and rows of nuclei (**c**; $n = 4$, Figures denote mean±SEM, unpaired two-tailed $t$-test, ** = $p < 0.01$). As in the *Rpgr*-KO mouse shown in Fig. 3, rhodopsin is mislocalised to the outer plexiform layer (*arrowhead*; **a**) and peri-nuclear area (*arrow*; **a**). **d** Increased GFAP immunolabeling throughout the radial length of Müller cells in the ONL is apparent (*arrow*) at 5 months of age. **e** Increased F-actin is seen in the connecting cilium (*arrowhead*) of *Gelsolin* KO mouse photoreceptor compared to wild-type. (Images representative of $n = 3$ analysed). Scale bars: 20 μm **a**, **e**

isoform could be explained by the finding that the two RPGR isoforms form a complex in vivo (WZ and HK, unpublished observations), whereupon the mutant RPGR^ORF15 prevents gelsolin binding to the RPGR^Ex1-19 isoform.

There are two potential mechanisms by which the dysregulation of actin caused by abnormalities of gelsolin activation could lead to the mislocalisation of rhodopsin, photoreceptor stress and eventual degeneration characteristic of XLRP (Fig. 7). First, by disturbing cilia formation, which impacts on the genesis and maintenance of the modified cilium that comprises the photoreceptor. It is well recognised that actin exerts an influence on ciliogenesis and cilia maintenance[19, 26–28]. Indeed, actin depolymerisation leads to lengthened, nascent, OS discs in photoreceptors[29, 30]. Actin is localised to the distal portion of the CC in human photoreceptors where the ciliary membrane evaginates to form OS discs[31–34]. These results suggest that RPGR facilitates OS disc budding or the completion of disc formation by locally regulating actin dynamics in a gelsolin-dependent manner. In keeping with this, the disorganised nature of discs reported in an *Rpgr* KO mouse[5] occurs with a hyperstabilised actin cytoskeleton in both mutant postnatal mouse eyes and *RPGR*-mutant photoreceptor organoids characterised here. Additional support comes from a role of RPGR in facilitating the OS content of inositol polyphosphatase INPP5E[35]. INPP5E regulates the phosphoinositide content of the membrane, which in turn regulates the local lipid-actin cytoskeleton interface[36]. Excitingly, recent work has demonstrated the highly localised role actin plays in excision of a membranous bud (ectosome) at the cilia tip[37, 38]. The process, known as ectocytosis (or decapitation), serves as an alternative ciliary exit route for G-protein-coupled receptors. We speculate that photoreceptor disc formation may share a common ancestral mechanism with ectocytosis and that RPGR's alternatively spliced ORF15 variant may have evolved to facilitate this.

Second, photoreceptor degeneration could occur if actin dysregulation functionally abrogates rhodopsin trafficking. In the photoreceptor, an actin bundle connects the periciliary membrane complex with the basal body at the base of the CC,

along which the actin-based motor protein myosin VIIA appears to travel[30, 39]. Myosin VIIa contributes to the active transport of visual pigments, including rhodopsin, into the CC[39]. Support for a role of RPGR and gelsolin in ciliary function and rhodopsin trafficking comes from work examining Usher Syndrome, a syndromic form of RP and deafness. This syndrome can be caused by *whirlin* (*WHRN*) mutations[40], coding for a protein that forms part of the usherin complex that regulates the actin filament network in the periciliary membrane complex[41, 42]. Gelsolin is part of this WHRN complex in inner ear stereocilia[18] where it is mislocalised away from stereocilia in the *whirlin* mutant mouse[18]. WHRN has also been shown to interact with the RPGR ORF15 basic domain in photoreceptors[43], in keeping with a model whereby loss of WHRN perturbs the regulation of gelsolin-mediated actin turnover by RPGR, disrupting myosin VIIa-mediated rhodopsin transport at the periciliary membrane complex. RPGR also facilitates the ciliary targeting of the small G protein, RAB8, which regulates vesicular trafficking during primary ciliogenesis and rhodopsin transport in the photoreceptor[42], supporting this model.

These models are not mutually exclusive, and further work will define their relative contributions to the pathology of XLRP, as well as addressing the important translational implication of our study that pharmacomodulation or ectopic expression of activated gelsolin in the retina could slow or reverse the retinal degeneration observed in *RPGR*/XLRP disease.

## Methods

**Antibodies and reagents**. Primary antibodies: mouse anti-OCT3/4 (R&D AF1759 10 μg/ml IF), goat anti-Nanog NL493-conjugated (R&D 967151; 1 in 10 IF), SOX2 (eBioscience 14-9811-82; 1 in 100 IF), rabbit anti-SMA (Abcam AB5694; 1 in 100 IF), β-Tub (Abcam ab7751; 1 in 1000 IF), mouse anti-αFP (Sigma A8452; 1 in 500 IF), rabbit anti-PAX6 (Covance PRB-278P; 1 in 200 IF), goat anti-CHX10 (Santa Cruz SC-21692; 1 in 500 IF), rabbit anti-RPGR (Atlas Antibodies HPA001593; 1 in 200 IF), rabbit anti-Recoverin (Chemicon AB5585; 1 in 100 IF), mouse anti-RETP1 (Chemicon MAB5326; 1 in 1000 IF), phalloidin-Alexa647 (Life Technologies a2287; 1 in 200 IF), Hoechst, rabbit anti-GFAP (DAKO ZO334; 1 in 100 IF), mouse anti-acetylated α-tubulin (Sigma, T6793; 1 in 200 IF), rabbit anti-Gelsolin (Abcam 74420; 1 in 900 WB), rabbit anti-Cofilin/pCofilin (Cell Signalling 3313 & 5175; 1 in 1000 WB), mouse anti-GAPDH (Chemicon MAB374; 1 in 1000 WB).

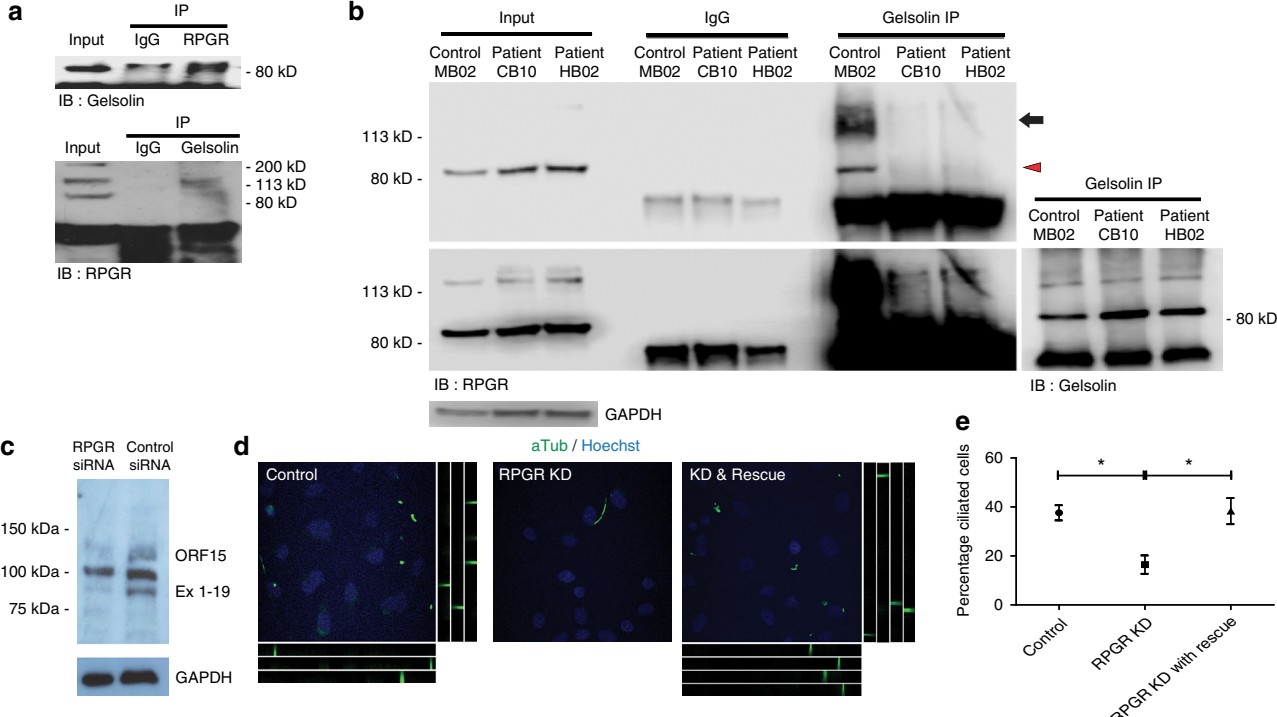

**Fig. 6** Loss-of-function and gain-of-function experiments support a role for gelsolin in *RPGR* mediated cilia maintenance. **a** Immunoprecipitation (IP) of RPGR (*upper* panel) and gelsolin (*lower* panel) demonstrates an endogenous RPGR-gelsolin interaction in bovine retina, as evidenced by immunoblotting (IB) showing the presence of gelsolin (86 kD) and the 127 kD retina-specific form of RPGR respectively in the immunoprecipitates. **b** Immunoprecipitation (IP) of gelsolin followed by immunoblotting (IB) for RPGR reveals that the RPGR-gelsolin interaction is present in control iPSC-derived photoreceptors but is perturbed in both XLRP patient-derived *RPGR*-mutant cultures—note in the righthand set of three lanes that RPGR is present only in the IPs from the control patient. The *lower* panel shows a longer exposure of the immunoblot to show that both the constitutively-expressed (90 kD, encoded by exons 1–19, arrowhead) and retina-specific (127 kD—encoded by exons 1–14 and specific open reading frame ORF 15, arrow) forms of RPGR are present in the control and mutant cell lysates shown in the three left hand lanes. *Far right* panel shows equal amounts of gelsolin are pulled down from each sample by gelsolin IP. **c** siRNA-mediated *RPGR* knockdown. Note that expression of both the constitutively-expressed (90 kD—Ex 1–19) and retina-specific (127 kD —ORF 15) forms of RPGR are reduced by the siRNA while the non-specific band at 100 kD is unaffected in the hTERT-RPE cell line. **d, e** Immmunolabeling of the hTERT-RPE cell line after siRNA-mediated *RPGR* knockdown reveals a loss of ciliogenesis (Mean percentage of ciliated cells 37.62%±3.113 vs 16.48% ±3.762 (SEM), unpaired two-tailed *t*-test, *p* = 0.0124; identified by acetylated α-tubulin immunoreactivity in green), as shown by a comparison of the control (*left*) and *RPGR* knockdown (*centre*) panels. The defect is rescued by overexpression of constitutively active gelsolin, as shown in the *right* panel (Mean percentage of ciliated cells 16.48%±3.762 vs 38.33%±5.349 (SEM), unpaired two-tailed *t*-test, *p* = 0.0288.) The tracks *below* and to the *right* of the micrographs in **d** show *z*-planes of the image to confirm the presence of elongated cilia. *Scale bars*: 10 µm (**d**)

Secondary antibodies (all Life Technologies; all 1 in 1000 dilution unless otherwise stated): donkey anti-mouse-Alexa488 (A21202), donkey anti-mouse-Alexa555 (A10037), donkey anti-goat-Alexa488 (A21432), donkey anti-goat-Alexa555 (A11055), donkey anti-sheep-Alexa488 (A11015), donkey anti-sheep-Alexa555 (A21099), donkey anti-rabbit-Alexa488 (A10042), donkey anti-rabbit-Alexa568 (A21206), Horseradish peroxidase rabbit (NA934V, GE Healthcare; 1 in 10000 WB) or Mouse (NA931V, GE Healthcare; 1 in 10,000 WB).

siRNA duplexes (all with dT overhangs) were purchased from Sigma-Aldrich: control siRNA: 5′-UUCUCCGAACGUGUCACGU-3′; *RPGR* siRNA: 5′-GAGAUAGAUAAUUCUUCAA-3′[6]. Constitutively active gelsolin (Ch-*gelsolN*) was a gift from Ikuo Wada (Addgene plasmid #37262)[24].

**iPSC generation.** Skin biopsies were performed under local anaesthetic (Xylocaine 2% with Adrenaline 100 micrograms/20 ml (1:200,000), AstraZeneca) using a sterile technique following ethical approval (REC Ref No 10/S1103/10, Lothian R&D Project No 2012/R/DER/01) and informed consent. Both patients and controls were aware of their use of their samples in research. Fibroblast cell lines developed in Dubecco's Modified Eagle Medium (Gibco) with 10% fetal calf serum (Gibco) and 1:500 Penicillin/Streptomycin (Gibco).

Three micrograms of expression plasmid mixtures (generous donation from Yamanaka lab)[9] were electroporated into 500,000 fibroblasts in Amaxa Nucleofector 2b (Lonza AAB1001) according to the manufacturer's instructions and transferred to a 6-well Gelatin-coated plate (1% Sigma) in DMEM/10% FCS. The cells were trypsinised upon reaching confluence and replated on Geltrek in Essential8 media[43] (E8; Gibco) in a 100 ml dish. Colonies were counted after 25 days and hESC-looking colonies picked and expanded. Cells were maintained on Geltrek in E8 and passaged with EDTA (0.5 M, Invitrogen).

**Photoreceptor differentiation.** Three-dimensional optic cup structures protocols were adapted from previous protocols[11, 12]. Confluent iPSC colonies were dissociated using EDTA (0.5 M, Invitrogen), resuspended in Essential6 media (Gibco) and allowed to form embryoid bodies in non-adherent 10 cm petri dishes (Sterilin) at 37 °C on an orbital shaker (Stuart). Twenty-four hours later they began patterning for 12 days with reducing knockout serum replacement (KOSR; 20% for 5 days, 15% until day 9, 10% until day 37), 1:50 B27 (PAA), 5 ng/ml rhIGF-1 (Sigma), 3 µm IWR1e and 1% Geltrek in DMEM/Ham's F-12 (Gibco). From day 12–18, cultures were incubated with reducing KOSR, 1:50 B27 (PAA), 5 ng/ml rhIGF-1 (Sigma), 10% FCS, 100 nm SAg and 1% Geltrek in DMEM/Ham's F-12 (Gibco). From day 18–37, DMEM: F-12 was supplemented with 10% KOSR, 1:50 B27 (PAA), 5 ng/ml rhIGF-1 (Sigma) and 1% Geltrek. From day 37 onwards, DMEM: F-12 was supplemented with 1:50 B27 (PAA), 1:100 N2 (Gibco), 10 ng/ml rhIGF-1 (Sigma) and 1% Geltrek.

**Animals.** All experiments were performed in compliance with the ARVO Statement for the Use of Animals in Ophthalmic and Vision Research, and the United Kingdom Animals (Scientific Procedures) Act 1986 under project licence number P8C815DC9 (Dr X Shu).

**Histology and immunostaining.** Mouse eyes[13, 17] were fixed immediately after enucleation and corneal excision in 4% paraformaldehyde (Sigma-Aldrich) in PBS at room temperature (RT) for 1 h, infiltrated with sucrose for cryoprotection (15%, 30%) and embedded in OCT (CellPath). Complete sectioning of whole eyes was performed through the horizontal meridian. Sections (8–50 µm thickness) were collected. Three-dimensional iPSC-derived optic cup structures were fixed,

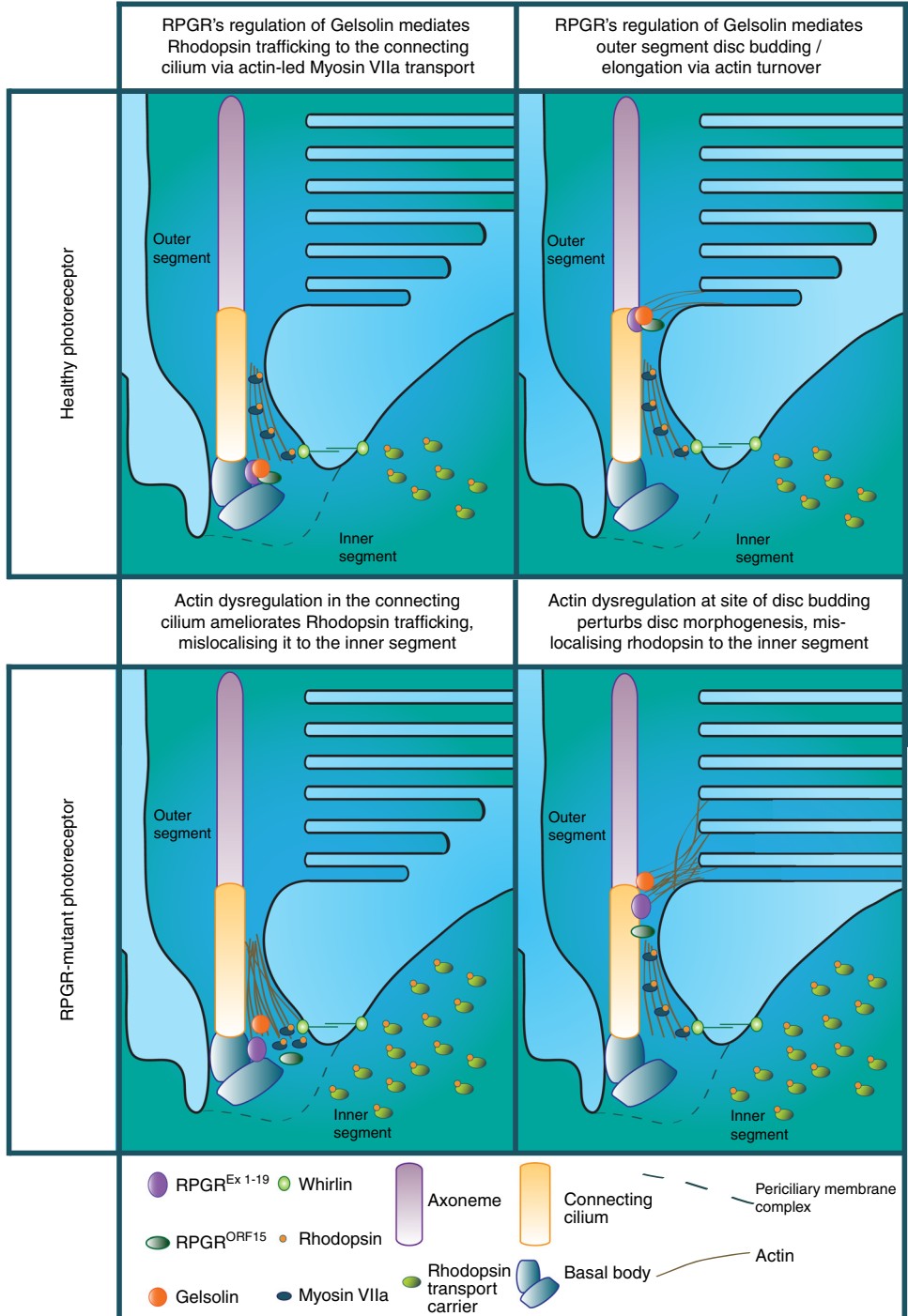

**Fig. 7** Proposed model for RPGR mediated actin turnover in the photoreceptor connecting cilium. We propose two possible mechanisms of action for RPGR. First, RPGR's regulation of actin turnover could mediate rhodopsin trafficking into the photoreceptor connecting cilium (*left* panels). An actin bundle connects the periciliary membrane complex to the basal body, along which Myosin VIIa actively transports visual pigments. Whirlin regulates this actin network at the connecting cilium base and interacts with both gelsolin and RPGR, suggesting a model whereby mutant RPGR disrupts the complex, perturbing actin turnover and compromising Myosin VIIa-mediated rhodopsin transport (*left* panels). The resulting rhodopsin mislocalisation to the inner segment results in cell stress and degeneration. Second, RPGR's regulation of actin turnover could facilitate outer segment disc formation (*right* panels). Actin exerts an influence on both ciliogenesis and maintenance and is localised to the site of photoreceptor disc budding. The disorganised discs and hyperstabilised actin seen in the Rpgr knockout mouse supports a model whereby RPGR facilitates outer segment disc budding or the completion of disc formation by gelsolin-mediated actin turnover (*right* panels). The compromised disc morphogenesis seen with mutant RPGR would mislocalise rhodopsin to the inner segment, resulting in cell stress and degeneration

cryoprotected, embedded and sectioned in a similar fashion. hTERT-RPE cells grown on slides were fixed in 4% paraformaldehyde for 30 min at RT.

Sections/cells were blocked/permeabilised with 2% NDS (Sigma), 2% BSA (Sigma) and 0.5% Triton X100 (Fisher) for 1 h at RT, washed with PBS, incubated with primary antibodies overnight at 4 °C, washed in PBS, incubated with secondary antibodies for 60 min at RT, washed in Phalloidin for 30 min at RT, washed in PBS, incubated in Hoechst for 5 min at RT and mounted with coverslips using Fluoromount-G (Southern Biotech). Images were taken on a Leica SPE confocal microscope and analysed using the ImageJ and ImagePro software. Images represent confocal projections, unless stated otherwise.

For actin levels, pixel intensity of phalloidin staining of standardised regions of interest (ROI) of confocal Z-stacks were measured by blinded assessors using ImageJ.

Assessors were blinded for all mouse retina measurements.

**Electron microscopy**. Three-dimensional optic cup structures were fixed in 2.5% glutaraldehyde (TAAB) in 0.1 M phosphate (PO) buffer at RT (2 h), washed (0.1 M PO buffer) and lipids stained with 1% osmium tetroxide (Electron Microscopy Science) (1 h). Samples were washed (PO buffer) prior to ethanol dehydration. Residual ethanol was removed with propylene oxide (TAAB) prior to overnight incubation in 5 g araldite CY212 (Agar Scientific) and 5 g dodecenyl succinic anhydride (DDSA, Agar Scientific). Sequential incubations in the above mix preceded transfer to final embedding resin (11.5 g araldite CY212, 11 g DDSA, 0.55 ml benzyldimethylamine (Agar Scientific), 0.5 ml dibutylphthalate (Agar Scientific). Repeated exchange of embedding resin preceded 48 h in an oven (60 °C).

**RNA extraction and RT-PCR**. mRNA was extracted from transfected cells using the RNeasy Minikit (Qiagen, Crawley, UK) before being reverse transcribed with the Superscript First Strand Synthesis System for RT-PCR (Invitrogen) as per manufacturer's instructions. The resulting cDNA was used to perform RT-PCR with the Roche Light Cycler 480 II using the QuantiFast SYBR Green PCR Kit (Qiagen) and the following primers: *L-MYC*: Fd 5′-CAGGGGGTCTGCTCG-CACCGTGATG-3′ & Rv 5′-TCAATTCTGTGCCTCCGGGAGCAGGGTAGG-3′; *OCT3/4*: Fd 5′-GTTGGAGAAGGTGGAACCAA-3′ & Rv 5′-CTCCTTCTGCA GGGCTTTC-3′; *SOX2*: Fd 5′-GTGTTTGCAAAAAGGGAAAAGT-3′ & Rv 5′-TCTTTCTCCCAGCCCTAGTCT-3′; *LIN28*: Fd 5′-AGCCATATGGTAGCC TCATGTCCGC-3′ & Rv 5′-TCAATTCTGTGCCTCCGGGAGCAGGG TAGG-3′; *PAX6*: Fd 5′-CTCGGTGGTGTCTTTGTCAAC-3′ & Rv 5′-ACTTTTGC ATCTGCATGGGTC-3′; *Rx*: Fd 5′-GAATCTCGAAATCTCAGCCC-3′ & Rv 5′-CTTCACTAATTTGCTCAGGAC-3′; *LHX2*: Fd 5′-CAAGATCTCGGACCG CTACT-3′ & Rv 5′-CGGTGGTCAGCATCTTGTTA-3′; *SIX3*: Fd 5′-CCGGAA-GAGTTGTCCATGTT-3′ & Rv 5′-CGACTCGTGTTTGTTGATGG-3′; *SIX6*: Fd 5′-ATTTGGGACGGCGAACAGAAGACA-3′ & Rv 5′-ATCCTGGATGGG-CAACTCAGATGT-3′; *CRX*: Fd 5′-GTGAGGAGGTGGCTCTGAAG-3′ & Rv 5′-CTGCTGTTTCTGCTGCTGTC-3′; *NESTIN*: Fd 5′-CAGGAGAAACAGGG CCTACA-3′ & Rv 5′-TGGGAGCAAAGATCCAAGAC5′-; *ARRESTIN*: Fd 5′-CTACCTGGGGAAACGGGACT-3′ & Rv 5′-GGCCATAGCGAAAGGC ACA-3′; *RECOVERIN*: Fd 5′-TTCAAGGAGTACGTCATCGCC-3′ & Rv 5′-GATGGTCCCGTTACCGTCC-3′

**Western blots**. Cells were lysed on ice in Pierce RIPA Buffer (ThermoScientific) containing Protease (Millipore) and Phosphatase (Calbiochem) inhibitors. Lysates were centrifuged (16000×*g*) for 10 min and the post-nuclear supernatants were incubated for 5 min at 95 °C with Laemmli sample buffer. Proteins were separated on a 4–20% Precise Tris-HEPES protein gel (ThermoScientific) and transferred to an Immobilon P Membrane (Merck Millipore). Blots were blocked in milk for 1 h before incubation with primary antibodies overnight in 4% BSA. A secondary antibody coupled with horseradish peroxidase was used for detection by a Licor C-Digit digital reader or a Konika Minolta SRX-101A developer with Super RX-N film (FujiFilm). All uncropped western blots can be found in Supplementary Fig. 7.

**Co-immunoprecipitation of bovine retina and iPSC cultures**. Bovine retina and photoreceptor cultures were lysed on ice with buffer (1% Triton-X100, 50 mMol Tris and 150 mMol NaCl in PBS (pH 7.4)) containing Protease (Millipore) and Phosphatase (Calbiochem) inhibitors. Lysates were centrifuged (16,000×*g*) for 10 min. Antibody was attached to Dynabeads Protein A (Life Technologies) and immunoprecipitation performed as per manufacturer's instructions. Supernatants were incubated for 10 min (70 °C) with Laemmli sample buffer. Proteins were separated on 4–20% precise Tris-HEPES protein gel (ThermoScientific) and transferred to an Immobilon P Membrane (Merck Millipore). Blots were blocked in milk and incubated with primary antibodies overnight in 4% BSA. Secondary antibody coupled with horseradish peroxidase was used for detection by a Licor C-Digit digital reader or a Konika Minolta SRX-101A developer with Super RX-N film (FujiFilm).

**Co-immunoprecipitation of recombinant proteins**. Cell Culture, Transient transfection and co-IP: hTERT-RPE1 cells were maintained in DMEM/F-12 (Invitrogen) plus 10% FBS and penicillin/streptomycin. For testing Gelsolin-RPGR interaction, we transiently transfected the cells with plasmids encoding FLAG-

tagged Gelsolin and GFP-tagged full length $RPGR^{Ex1-19}$ (const), $RPGR^{ORF15}$ and different RPGR domains ($RPGR^{1-11}$, $RPGR^{1-15}$, $RPGR^{12-15}$ or $RPGR^{16-19}$). Details of these constructs have been described previously. At the end of 48 h, cells were lysed in IP lysis buffer [25 mM Tris pH 7.4, 150 mM NaCl, 1 mM EDTA, 1% NP-40, 5% Glycerol and Complete Protease Inhibitor (Roche)]. The lysates were centrifuged to remove the debris and subjected to pulldown using GFP antibody crosslinked to AminoLink Plus Coupling Resin. The details of crosslinking have been described[35]. The immunoprecipitate was washed three times with the IP lysis buffer and the samples were eluted in glycine lysis buffer (pH 2.8), neutralised using 1 M Tris pH 9.5 and analysed by SDS-PAGE and immunoblotting.

**Gelsolin assay for determination of triton-insoluble F-actin**. Cell cultures were lysed with buffer containing 120 mM PIPES, 50 mM HEPES, 4 mM MgCl₂, 10 mM glucose, 20 mM EDTA, 0.1 mM dithiothreitol, 1 mM phenylmethylsulfonylfluoride and protease (Millipore) and phosphatase (Calbiochem) inhibitors. Initial lysate also contained 1.5% Triton-X100 to isolate Triton-soluble G-Actin (inactive/unbound Gelsolin)[22, 23]. Lysate was vortexed (10 min) and spun (12,000×*g*) for 10 min, all at 4 °C. Triton-soluble supernatant (G-actin; inactive/unbound gelsolin) was removed and the pellet resuspended in lysis buffer containing 6% SDS prior to 10 min vortexing and centrifugation (12000×*g*) for 10 min, all at RT. Resulting supernatant contained F-actin (active/bound gelsolin). Lysates were then processed for western blotting.

**Phospho protein arrays**. Arrays were carried out as per manufacturer's instructions (Fullmoon Bio). Briefly, 100 day old photoreceptor cultures (RPGR-mutant vs Control, *n* = 1) were lysed with extraction buffer containing Protease (Millipore) and Phosphatase (Calbiochem) inhibitors, purified and biotinylated. Array chips were blocked and biotinylated proteins allowed to couple to chips. Chips were washed, incubated with Cy3 streptavidin, washed and measured on Axon 4200 A slide scanner.

**Transfection and cilia formation**. Cells were transfected with Lipofectamine2000 (Invitrogen) using the reverse transfection protocol as per manufacturer's instructions. 0.125 µl control/*RPGR* siRNA[1] and 1 µl Lipofectamine2000 were diluted in 100 µl OptiMEM (Gibco) and incubated for 20 min in a 24-well plate at RT. Cells were plated on the siRNA/Lipofectamine mix at $3 \times 10^4$ cells/well. Twenty-four hours later the siRNA/Lipofectamine mix was removed and replaced by 2 µl *Ch-gelsolN*[24], 2 µl Lipofectamine and 100 µl OptiMEM (preincubated for 20 min at RT). Four hours later, the medium was replaced with serum-free medium for 24 h, after which cells were fixed, immunostained and assessed for cilia formation.

**Data availability**. All data generated or analysed during this study are included in this published article (and its Supplementary Information) or from the corresponding author upon reasonable request.

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

## Acknowledgements

We would like to acknowledge Dr Xinhua Shu (Glasgow Caledonian University) for kindly housing the *Rpgr* KO mice. We would also like to thank CB, HB, MB and KR for their generous tissue donation to generate the iPSCs. This work was supported by grants from the Wellcome Trust (R.M., H.A. by Grant Number 100470/Z/12/Z; C.f.-C. by an Investigator Award), Retinitis Pigmentosa Fighting Blindness (R.M., H.A., A.F.W.; Grant Number GR583), the Academy of Medical Sciences (R.M., M.J.; Grant Number SGL014\1011), a Medical Research Council Programme Grant (A.F.W.), an ERC Fellowship (C.M., M.L.), a DFG Grant (C.G., W.W.; Grant Number SPP1464) and the NIH (W.Z., H.K.; Grant Number EY022372).

## Author contributions

R.M., A.F.W., M.L. and C.f.-C. designed research; R.M., H.A.-A., M.J., C.G. and W.Z. performed research. R.M., H.A-A., M.J., C.M., H.K., P.M., M.L., C.f.-C. analysed data; R. M. and C.f.-C. wrote the manuscript; R.M., C.M., W.W., H.K., P.M., B.D. A.F.W., M.L. and C.f.-C. edited the manuscript.

## Additional information

**Competing interests:** The authors declare no competing financial interests

