## [Peer Review File · Nature Communications]

Reviewers' Comments:

Reviewer #1 (Remarks to the Author)

In the present manuscript, Megaw and colleagues found that RPGR/XLRP mutation leads to increased actin polymerization in differentiating photoreceptors using hiPSC-derived retinal tissue. This is supported by data that in RPGR/XLRP photoreceptor culture cofilin proteins are more phosphorylated than that in retinal tissue from control hiPCS. Furthermore, they demonstrate that RPGR mutations perturb the RPGR-gelsolin interaction and leads to downregulation of gelsolin activity. These findings are confirmed by retinal phenotypes of RPGR and gelsolin mutant mice, respectively. The manuscript is well written and experiments were performed well. However, there are few points of major criticism that need to be addressed before the manuscript should be considered for publication.

Major concerns:

1, There is only one example of organoids displaying characteristic phenotype of photoreceptor. How reproducible is this phenotype? The authors should show a range of images displaying the variation in defect of photoreceptors. As well as that, authors show only high magnification of view of retinal tissue. Lower magnification images are also required to clearly show that actin polymerization was increased in the hiPSC-derived differentiating photoreceptors.

2, In figure 3g, increase in phalloidin signaling in photoreceptors is not phenocopied in RPGR mutant mice. This may be caused by the difference in the developing stages. hiPS-derived photoreceptor cultured for 100 days is not matured one, that is still embryonic stage. So, authors should examined phalloidin staining in RPGR mutant mice at more early stage.

3, It is not clear how quantify the intensity of phalloidin immunostaining. Fluorescent intensity was carefully compared with any internal control signals.

4, Authors show that phosphorylated cofilin proteins are increased in RPGR/XLRP retinal tissues by western blot. But, it is not clear where are phospho-cofilins localized in developing photoreceptors, and whether they are indeed increased in RPGR/XLRP photoreceptors? To clarify these, they should also show the results of phospho-cofilin immunostaining in hiPSC-derived retinal tissues (control and patient derived), gelsolin mutant mice and rpgr mutant mice, respectively.

5, In figure 6d, authors say that over expression of active gelsolin could rescue RPGR-loss phenotype in ciliogenesis. Although this result is directly support their claim that gelsolin is functionally related with RPGR, they only show few images from each conditions. To eliminate the possibility of artifacts, quantification analysis of ciliogenesis is also required.

Reviewer #2 (Remarks to the Author)

My review is focused on the use of phospho-arrays. The data shown in Fig. 4 appear to support the conclusion that cortacin is hyperphosphorylated in the absence of RPGR.

The subsequent work on gelsolin being a partner of RPGR and excessive actin polymerization being responsible for retinal degeneration is an interesting hypothesis that will stimulate discussion in the field.

Reviewer #3 (Remarks to the Author)

In the manuscript entitled "Gelsolin dysregulation causes photoreceptor loss in induced pluripotent stem cell and animal models of retinitis pigmentosa" Megaw et al. propose that impaired F-actin disassembly in photoreceptors carrying mutations in the ciliopathy gene RPGR could be the basis

for rhodopsin mislocalization and subsequent photoreceptor cell death. The authors showed that photoreceptor (PR) differentiated from iPSCs from patients carrying a similar RPGR mutation as well as PRs from a RPGR-KO mouse model exhibit increased actin polymerization. At the molecular level, they identified an increase in phosphorylation of cofilin in RPGR mutant which is known to bind to and promote actin disassembly in its unphosphorylated state. Since phosphorylation of cofilin is known to be regulated, among other proteins, by actin-severing protein gelsolin, the authors further examined the photoreceptor morphology in a gelsolin knockout mouse and showed mislocalization of rhodopsin, loss of outer nuclear layer (ONL) thickness, and disruption of outer segments (OS) similar to that observed in the RPGR-KO mouse. Co-immunoprecipitation studies revealed that gelsolin interacts with WT but not mutant RPGR. Finally, the authors claim restoration of cilia formation in an RPGR-depleted RPE cell line by expression of a constitutively active form of gelsolin. The authors conclude that under WT conditions RPGR activates gelsolin in photoreceptors and thus, is involved in regulation of actin dynamics. This is the first paper describing the interaction of RPGR with gelsolin and proposes a novel possible pathogenic mechanism linking RPGR mutations with rhodopsin mislocalization. While it does not explain the molecular events that subsequently trigger PR cell death, it adds to our understanding of changes in protein trafficking in the connecting cilium of mutant RPGR/XLRP rods and may also contribute towards better explaining other retinal ciliopathies.

The following comments/suggestions are proposed that would significantly strengthen the value of this paper.

Major comments:

1. Statistical tests used have neither been described in the Material & Methods section nor in the figure legends (Fig 2, Fig3, Fig 5). What statistical parameter of central tendency (mean, median) was used? What do the error bars show ({plus minus}SD, {plus minus}2SD, {plus minus}SEM, {plus minus}2SEM)?

2. The manuscript falls short of identifying the binding site of Gelsolin on RPGR. Is there anything that can be deduced from the Gelsolin co-IP result shown in Figure 6? How do you reconcile the absence of both the constitutive RPGR1-19 and RPGR1-ORF15 bands from the patient derived RPGR mutant cultures? Well-conceived experiments (e.g. co IP studies or yeast two hybrid assays) utilizing RPGR variants that have deleted regions should be included to narrow down the region of interaction between these 2 proteins. Such data may inform as to whether the proposed pathogenic mechanism (and potential gelsolin augmentation therapeutic strategy) would only apply to the subset of RPGR mutations that alter this binding region.

3. The authors have shown some preliminary evidence supporting the claim that overexpression of gelsolin can correct the absence of ciliogenesis (Fig 6d). However there are some limitations to the current results:

a) The conclusion is purely based on qualitative results showing some α -tubulin labeled structures.

The authors should consider providing quantitative data AND electron microscopy imaging to unequivocally state that a cilium (with its microtubule doublets organization) is being formed.

b) The cell used in this study is not a photoreceptor but an RPE cell line. As the authors have access to the RPGR KO mouse why not consider demonstrating whether gelsolin gene augmentation corrects RHO mislocalization and the RPGR-induced cell death process?

4. A well-described feature of RPGR disease is (as reported by the authors) rhodopsin mislocalization, however mislocalization of cone opsins has also been found in mice, dogs, and humans carrying RPGR mutations. Because of the importance of cone-mediated visual impairment in this condition, and current efforts at therapeutically addressing the defect also in this PR cell population, it seems as an oversight to not have investigated in the gelsolin KO mouse whether cone opsin mislocalization is also seen. Regardless of the results, this would be very informative,

and these findings should be discussed.

Minor comments:

a) The title of the paper does not truly capture the conclusions that can be drawn from the results of this paper. While the studies show an interaction between RPGR and gelsolin, there is no experiment demonstrating a "dysregulation" of gelsolin or even establishing gelsolin dysregulation as a direct "cause" for PR cell loss. It is recommended that the title be changed to better reflect the conclusions that can be drawn based on the results presented in the paper.

b) The order of the panels from Fig 1 is confusing and does not follow the flow of the description in the Result section. Please reorder or rename.

c) Could the authors please address the discrepancy in the localization of recoverin (Fig.2a, used as a marker for photoreceptor differentiation) in patient derived iPSC cell lines? Is there any evidence for recoverin mislocalization in RPGR/XLRP photoreceptors? Have the authors observed differences in recoverin immunostaining between normal and RPGR-KO mouse photoreceptors?

d) Under the Results section describing the data presented under Fig 2b, the authors claim that "The results show that cell signaling pathways that regulate actin turnover in the photoreceptor are disrupted by RPGR/XLRP mutations". No such evidence is at all presented. Which are those specific signaling pathways? The results from the 1st assay on the 639 phosphoproteins should be fully disclosed in the manuscript possibly as Supplementary Information, and specific pathways thought to act on actin turnover clearly identified. In addition, the authors should indicate in the Methods section what is the source of the phosphoprotein arrays used (indicate company and catalog number).

e) In the result section, describing the increased GFAP labeling seen in Fig 3 and Fig 5 a, the authors indicate that Müller cells (not Muller cells) are activated and have migrated. The reviewer agrees that increase in GFAP immunolabeling throughout the radial length of a Müller is an unspecific sign of reactivity (ie gliosis), but disagrees with the description of cell migration. There is no evidence whatsoever from the images shown that the cell body of the Müller cells have migrated from the INL to the ONL.

f) For figures 3b, 3c and 3h, the asterisks indicate significant change when compared to which value? Please use brackets to indicate this as used for figures 2b, 5b and 5c.

g) In the RPGR-KO mouse photoreceptors, there is an increase in the length of actin fibers but a decrease in the phalloidin staining intensity (Fig.3g). What is a possible explanation for this?

h) The result title "RPGR activates the actin-severing protein Gelsolin" is again misleading. The results only show decreased binding of gelsolin to F-actin in RPGR/XLRP; it does not establish RPGR mutation by itself to be a cause of this decrease and certainly does not support the direct involvement of RPGR in activating gelsolin.

i) In Figure 4a, there is a slight discrepancy between the fold-change value for ser3 phosphorylation on cofilin stated by the authors in the text (3.16 fold change; RPGR activates the actin-severing protein Gelsolin, line 4) and that indicated by the color key in the figure. Based on the figure 4a, there appears to be higher than 4-fold change in the cofilin phosphorylation.

j) As the authors claim that the "open" Gelsolin which is the form bound to F-actin is decreased in RPGR mutant iPSC-derived PRs, the authors should include in Fig 4C an actin immunoblot to normalize against the amount of F-actin obtained from each sample. It is not clear why they are

showing instead GAPDH.

k) It would be informative to calculate the change in actin fiber length in the photoreceptors of the gelsolin-KO mice (Fig. 5), similar to the measurements made in the RPGR-KO photoreceptors (Fig.3g).

l) In the co-IP immunoblots using gelsolin IP (Fig.6b), the authors should include IB for gelsolin in order to show whether gelsolin was immunoprecipitated at equivalent amounts in the control and patient cell samples. They should also indicate the MW of gelsolin in the legend of Fig 6.

m) While the gelsolin misregulation could possibly explain the cofilin hyper-phosphorylation, a number of other cytoskeletal-modifying proteins are also hyper-phosphorylated in the RPGR/XLRP iPSC-derived photoreceptors. The authors should experimentally prove that it is indeed a disruption of gelsolin-cofilin interaction that is leading to hyper-phosphorylation of cofilin. Is cofilin hyper-phosphorylated in the gelsolin-KO photoreceptors or in gelsolin depleted RPE as well? And does expression of activated gelsolin in RPE alter cofilin phosphorylation in these cells? -

n) A final diagram, either in the main paper or in the supplementary material, with a model explaining the possible role of RPGR-gelsolin binding and downstream molecular interactions will be a useful addition to clearly explain the conclusions.

o) Both the introduction and discussion sections are particularly brief. The authors should consider discussing (or addressing) the limitations of this study, and also refer to other published work that has proposed other roles for RPGR (e.g. Murga-Zamalloa CA et al. Hum Mol Genet 2010; 19:3591; Murga-Zamalloa CA et al. Mol Vis 2010, 16:1373; Wright RN et al. IOVS 2011, 52:5189).

p) References 18 and 19 are incorrectly cited and seem irrelevant to the statements made by the authors. Please revise.

| Dear Referees

Many thanks for your comments and feedback on our article. You have suggested further experiments and revisions to enhance the body of work. Here we describe these experiments and revisions in turn, beginning with Reviewer #1.

Reviewer #1:

(#1) There is only one example of organoids displaying characteristic phenotype of photoreceptor. How reproducible is this phenotype? The authors should show a range of images displaying the variation in defect of photoreceptors. As well as that, authors show only high magnification of view of retinal tissue. Lower magnification images are also required to clearly show that actin polymerization was increased in the hiPSC-derived differentiating photoreceptors.

RESPONSE #1

We have included, as a supplementary figure (Supplementary Figure 3), further lower magnification images of control and patient iPSC-derived photoreceptor cultures which demonstrate the increased phalloidin staining in recoverin-positive cells in RPGR-mutant lines.

(#2) In figure 3g, increase in phalloidin signaling in photoreceptors is not phenocopied in RPGR mutant mice. This may be caused by the difference in the developing stages. hiPS-derived photoreceptor cultured for 100 days is not matured one, that is still embryonic stage. So, authors should examine phalloidin staining in RPGR mutant mice at more early stage.

RESPONSE #2

We believe that the increase in phalloidin staining seen in human iPSC-derived photoreceptors was in fact phenocopied in our *Rpgr*-mutant mice, with increased actin in the connecting cilium (Fig 3g). As the reviewer suggests, we have now examined the *Rpgr*-mutant mice in development (P2 and P10; new Supplementary Figure 4). We found that no difference in the photoreceptor actin cytoskeleton was seen in the P2 or P10 *Rpgr* KO mouse retina when compared to control. Thus, *Rpgr*'s role in actin regulation does not appear during these developmental stages *in vivo* – indeed we suggest that it does not appear until after eye opening at P14. We propose that the accelerated actin phenotype seen in hiPSC-derived photoreceptor cultures reflects the stress the organoids are under in the artificial *in vitro* environment, just as iPSC cells from individuals with late onset neurodegenerative diseases such as MND reveal phenotypes much earlier than *in vivo*.

(#3) It is not clear how quantify the intensity of phalloidin immunostaining. Fluorescent intensity was carefully compared with any internal control signals.

RESPONSE #3

Examining the same regions of interest on our images as those used for the quantification of phalloidin immunostaining, and using the same parameters, we compared the intensity of nuclei staining (Hoechst 5µg/ml) as an internal control. We found no difference in the intensity of Hoechst staining in our RPGR-mutant iPSC derived photoreceptors compared to control iPSC-derived photoreceptors (data not shown). Given this internal control, we conclude that the difference seen in phalloidin staining between RPGR-mutant and control iPSC-derived photoreceptor cultures is significant.

*(#4) Authors show that phosphorylated cofilin proteins are increased in RPGR/XLRP retinal tissues by western blot. But, it is not clear where are phospho-cofilins localized in developing photoreceptors, and whether they are indeed increased in RPGR/XLRP photoreceptors? To clarify these, they should also show the results of phospho-cofilin immunostaining in hiPSC-derived retinal tissues (control and patient derived), gelsolin mutant mice and *rpgr* mutant mice, respectively.*

RESPONSE #4

This is an excellent suggestion, but unfortunately not one we could address as no reliable commercial antibody exists for immunostaining of phospho-cofilin. We have however looked at the overall levels of cofilin expression in gelsolin mutant mice and *Rpgr* mutant mice. There is strong staining in the photoreceptor layer of the wild type retina. There does not appear to

be any difference in protein localization between wild type and mutant retinas / cultures (Supplementary Figure 5), although, the degenerate nature of the retina means the outer segments are not as well defined.

(#5) *5. In figure 6d, authors say that over expression of active gelsolin could rescue RPGR-loss phenotype in ciliogenesis. Although this result is directly support their claim that gelsolin is functionally related with RPGR, they only show few images from each conditions. To eliminate the possibility of artifacts, quantification analysis of ciliogenesis is also required.*

RESPONSE #5

We have added to figure 6 the graphs (6e) containing the quantification of rescue by constitutively active gelsolin of the ciliogenesis defect seen when RPGR is knocked down in the RPE cell line. The means, SEM and p values are now documented in the legend of figure 6.

Reviewer #2:

(#1) *My review is focused on the use of phospho-arrays. The data shown in Fig. 4 appear to support the conclusion that cortacin is hyperphosphorylated in the absence of RPGR. The subsequent work on gelsolin being a partner of RPGR and excessive actin polymerization being responsible for retinal degeneration is an interesting hypothesis that will stimulate discussion in the field.*

RESPONSE #1

We were pleased that Reviewer #2 felt the article to be interesting and note that he/she had not suggested we perform any further experiments.

Reviewer #3:

(#1) *Statistical tests used have neither been described in the Material & Methods section nor in the figure legends (Fig 2, Fig3, Fig 5). What statistical parameter of central tendency (mean, median) was used? What do the error bars show ({plus minus}SD, {plus minus}2SD, {plus minus}SEM, {plus minus}2SEM)?*

RESPONSE #1

We have amended all our figure legends to include the statistical information required (Mean +/-SEM)

(#2) *The manuscript falls short of identifying the binding site of Gelsolin on RPGR. Is there anything that can be deduced from the Gelsolin co-IP result shown in Figure 6? How do you reconcile the absence of both the constitutive RPGR1-19 and RPGR1-ORF15 bands from the patient derived RPGR mutant cultures? Well-conceived experiments (e.g. co IP studies or yeast two hybrid assays) utilizing RPGR variants that have deleted regions should be included to narrow down the region of interaction between these 2 proteins. Such data may inform as to whether the proposed pathogenic mechanism (and potential gelsolin augmentation therapeutic strategy) would only apply to the subset of RPGR mutations that alter this binding region.*

RESPONSE #2

We agree that defining the binding site for gelsolin on RPGR is important, and like the reviewer we were also struck by the observation that both the constitutive RPGR1-19 and RPGR1-ORF15 forms of RPGR are missing from the gelsolin co-IP in the mutant cells. To address these issues we collaborated with Hemant Khanna at the University of Massachusetts. Using his established RPGR Co-IP profile, we found an RPGR-Gelsolin interaction in the constitutive splice variant RPGR1-19 but not in RPGR1-ORF15 (Supplementary Figure 6). This suggests that the Gelsolin interacting domain of RPGR is within the basic domain encoded by the C terminal region of RPGR1-19. The lack of binding between RPGR1-ORF15 and Gelsolin is at first sight confusing, but based on preliminary data from the Khanna lab that the two forms of RPGR form a complex we hypothesize in the discussion that this complex is required for gelsolin binding and that the XLRP mutation that leads to a disordered ORF15 prevents the formation of this complex and so prevents gelsolin binding and activation.

(#3) *The authors have shown some preliminary evidence supporting the claim that*

overexpression of gelsolin can correct the absence of ciliogenesis (Fig 6d). However there are some limitations to the current results:

a) The conclusion is purely based on qualitative results showing some α -tubulin labeled structures. The authors should consider providing quantitative data AND electron microscopy imaging to unequivocally state that a cilium (with its microtubule doublets organization) is being formed.

b) The cell used in this study is not a photoreceptor but an RPE cell line. As the authors have access to the RPGR KO mouse why not consider demonstrating whether gelsolin gene augmentation corrects RHO mislocalization and the RPGR-induced cell death process?

RESPONSE #3

Please see our response to Reviewer #1's comment #5. We have added to figure 6 the graphs (6e) containing the quantification of rescue by constitutively active gelsolin of the ciliogenesis defect seen when RPGR is knocked down in the RPE cell line. The means, SEM and p values are documented in the legend of figure 6.

Reviewer #3 has also asked for confirmation of cilia formation by electron microscopy. Given that hTERT-RPE are ciliated cells, we could certainly use EM to produce a picture of a cilium in our cultures. We are not sure however what this would add. EM is not a quantitative technique and so we instead have used a standard cilia marker and 3 dimensional imaging (in the z plane) to quantify cilia.

Finally, Reviewer #3 has asked for demonstration of rescue of the photoreceptor degeneration seen in the Rprgr KO mouse by using gene augmentation. We agree that set of translational experiments are the next logical step for the project. They will not only confirm our proposed mechanism but also offer a possible novel therapeutic for RPGR/XLRP and have amended our discussion to this end. However, we see these experiments as a very substantial undertaking and out with the scope of this paper.

(#4) A well-described feature of RPGR disease is (as reported by the authors) rhodopsin mislocalization, however mislocalization of cone opsins has also been found in mice, dogs, and humans carrying RPGR mutations. Because of the importance of cone-mediated visual impairment in this condition, and current efforts at therapeutically addressing the defect also in this PR cell population, it seems as an oversight to not have investigated in the gelsolin KO mouse whether cone opsin mislocalization is also seen. Regardless of the results, this would be very informative, and these findings should be discussed.

RESPONSE #4

Thank you for suggesting this extension to our study. We have carried out the suggested experiment and report that there is no cone opsin mislocalisation in the Gelsolin KO mouse (this data is not included in the manuscript but provided as a figure for the reviewer). This interesting results illustrates key differences between rod and cone biology in the regulation of opsin localization.

***Minor Comment a** The title of the paper does not truly capture the conclusions that can be drawn from the results of this paper. While the studies show an interaction between RPGR and gelsolin, there is no experiment demonstrating a "dysregulation" of gelsolin or even establishing gelsolin dysregulation as a direct "cause" for PR cell loss. It is recommended that the title be changed to better reflect the conclusions that can be drawn based on the results presented in the paper.*

RESPONSE a

The reduced RPGR-gelsolin interaction in RPGR-mutant photoreceptor cultures, together with western blot demonstration of reduced Gsn activation, evidences the dysregulation of gelsolin. The retinal degeneration seen in the Gsn ko mouse evidences gelsolin loss as a direct cause for photoreceptor cell loss. We presume that the reviewer is highlighting the difference between a failure of gelsolin activation and the complete absence of gelsolin as reason for caution in assuming causality. However, given that active gelsolin is required for F-actin cleavage, we argue that the two can be equated. In light of the reviewer's comments, however, we have tried to be more precise in a new title "Loss of gelsolin function as a cause of photoreceptor loss in induced pluripotent cell- and animal models of retinitis pigmentosa"

***Minor Comment b** The order of the panels from Fig 1 is confusing and does not follow the flow of the description in the Result section. Please reorder or rename.*

RESPONSE b

We have reordered Fig 1, placing the longitudinal and cross-sectional electron microscopy images of the connecting cilium as 1g and 1h respectively. We have moved the immunostaining demonstrating the cilia localization of RPGR to 1i and the image of membranous material to 1j. We hope the reviewer feels this now improves the flow of the description of our photoreceptor cultures.

Minor Comment c *Could the authors please address the discrepancy in the localization of recoverin (Fig. 2a, used as a marker for photoreceptor differentiation) in patient derived iPSC cell lines? Is there any evidence for recoverin mislocalization in RPGR/XLRP photoreceptors? Have the authors observed differences in recoverin immunostaining between normal and RPGR-KO mouse photoreceptors?*

RESPONSE c

Whilst some iPSC-derived photoreceptors exhibit differences in recoverin staining/localization, the majority do not (see Supplementary Figure 3). To confirm that RPGR plays no role in recoverin transport, we looked at localization in the knock out mouse and found no difference between it and wild type retina (this data is not included in the manuscript but provided as a figure for the reviewer).

Minor Comment d *Under the Results section describing the data presented under Fig 2b, the authors claim that "The results show that cell signaling pathways that regulate actin turnover in the photoreceptor are disrupted by RPGR/XLRP mutations". No such evidence is at all presented. Which are those specific signaling pathways? The results from the 1st assay on the 639 phosphoproteins should be fully disclosed in the manuscript possibly as Supplementary Information, and specific pathways thought to act on actin turnover clearly identified. In addition, the authors should indicate in the Methods section what is the source of the phosphoprotein arrays used (indicate company and catalog number).*

RESPONSE d

We have now included the full results from the first assay as Supplementary Table 1. Further, we have listed the source of the phosphoprotein arrays (Fullmoon Bio) in our Methods section.

Minor Comment e *In the result section, describing the increased GFAP labeling seen in Fig 3 and Fig 5 a, the authors indicate that Müller cells (not Muller cells) are activated and have migrated. The reviewer agrees that increase in GFAP immunolabeling throughout the radial length of a Müller is an unspecific sign of reactivity (ie gliosis), but disagrees with the description of cell migration. There is no evidence whatsoever from the images shown that the cell body of the Müller cells have migrated from the INL to the ONL.*

RESPONSE e

We agree, and have amended our references to Müller cell activation in both the text and figures 3 and 5. We have removed mention of Müller cell migration and instead documented increased GFAP labeling, throughout the radial length of the cells. We have also added a diaeresis/umlaut to the ü of all Müller mentioned in the text – many apologies for this typographical oversight.

Minor Comment f *For figures 3b, 3c and 3h, the asterisks indicate significant change when compared to which value? Please use brackets to indicate this as used for figures 2b, 5b and 5c.*

RESPONSE f

The asterisks in figures 3b, 3c and 3h indicate significant change when compared to the 4 months old wild type mouse. We have amended the figure 3 legend to indicate this.

Minor Comment g *In the RPGR-KO mouse photoreceptors, there is an increase in the length of actin fibers but a decrease in the phalloidin staining intensity (Fig. 3g). What is a possible explanation for this?*

RESPONSE g

The use of actin fibre length within the inner segment as an assay for photoreceptor pathology has been removed from the revised version of the paper. We now believe (from the additional experiments above) that RPGR and gelsolin form a complex. As the goal is to measure the effect on F-actin levels that results from loss of either RPGR or gelsolin from this complex, the analysis is now focused only on the region where both are normally co-

expressed – the connecting cilium.

Minor Comment h *The result title "RPGR activates the actin-severing protein Gelsolin" is again misleading. The results only show decreased binding of gelsolin to F-actin in RPGR/XLRP; it does not establish RPGR mutation by itself to be a cause of this decrease and certainly does not support the direct involvement of RPGR in activating gelsolin.*

RESPONSE h

We accept this point, and have retitled this results section 'RPGR mutations ameliorates the activation of the actin-severing protein Gelsolin.'

Minor Comment i *In Figure 4a, there is a slight discrepancy between the fold-change value for ser3 phosphorylation on cofilin stated by the authors in the text (3.16 fold change; RPGR activates the actin-severing protein Gelsolin, line 4) and that indicated by the color key in the figure. Based on the figure 4a, there appears to be higher than 4-fold change in the cofilin phosphorylation.*

RESPONSE i We have remade the heat map for figure 4, with Cofilin's fold change (3.16) more accurately depicted by the updated scale bar

Minor Comment j *As the authors claim that the "open" Gelsolin which is the form bound to F-actin is decreased in RPGR mutant iPSC-derived PRs, the authors should include in Fig 4C an actin immunoblot to normalize against the amount of F-actin obtained from each sample. It is not clear why they are showing instead GAPDH.*

RESPONSE j

We have added a supplementary figure (Supplementary Figure 3) showing that activated gelsolin is present in reduced amounts in RPGR-mutant photoreceptor cultures when the western blots are normalized to F-actin levels.

Minor Comment k *It would be informative to calculate the change in actin fiber length in the photoreceptors of the gelsolin-KO mice (Fig. 5), similar to the measurements made in the RPGR-KO photoreceptors (Fig.3g).*

RESPONSE k

Please see our response to Minor Comment g above.

Minor Comment l *In the co-IP immunoblots using gelsolin IP (Fig.6b), the authors should include IB for gelsolin in order to show whether gelsolin was immunoprecipitated at equivalent amounts in the control and patient cell samples. They should also indicate the MW of gelsolin in the legend of Fig 6.*

RESPONSE l

We have included the MW of gelsolin (86kD) in the legend of Fig 6. We have also included in the image, as requested, an IB for gelsolin demonstrating the protein was immunoprecipitated at equivalent amounts in the control and patient cell samples.

Minor Comment m *While the gelsolin misregulation could possibly explain the cofilin hyper-phosphorylation, a number of other cytoskeletal-modifying proteins are also hyper-phosphorylated in the RPGR/XLRP iPSC-derived photoreceptors. The authors should experimentally prove that it is indeed a disruption of gelsolin-cofilin interaction that is leading to hyper-phosphorylation of cofilin. Is cofilin hyper-phosphorylated in the gelsolin-KO photoreceptors or in gelsolin depleted RPE as well? And does expression of activated gelsolin in RPE alter cofilin phosphorylation in these cells? –*

RESPONSE m

We have carried out the experiment suggested by the reviewer and found that Cofilin is not hyperphosphorylated in the Gelsolin KO mouse. However, it is important to note that we are not arguing that the hyperphosphorylation of cofilin has a causal role in the photoreceptor pathology of our models. Rather, it was significant within the context of our study simply because it pointed to the role of gelsolin, and abnormalities of its activation, in these models. We would suggest that the alterations within feedback loops following complete loss of gelsolin (the knock out) likely differ from those following perturbation of activation (iPSC cultures, and this explains the differing levels of cofilin phosphorylation observed.

Minor Comment n *A final diagram, either in the main paper or in the supplementary material,*

with a model explaining the possible role of RPGR-gelsolin binding and downstream molecular interactions will be a useful addition to clearly explain the conclusions.

RESPONSE n

We have made a diagram (new Supplementary Figure 7) that depicts our proposed model for RPGR's role in connecting cilium actin turnover and rhodopsin trafficking.

Minor Comment o *Both the introduction and discussion sections are particularly brief. The authors should consider discussing (or addressing) the limitations of this study, and also refer to other published work that has proposed other roles for RPGR (e.g. Murga-Zamalloa CA et al. Hum Mol Genet 2010; 19:3591; Murga-Zamalloa CA et al. Mol Vis 2010, 16:1373; Wright RN et al. IOVS 2011, 52:5189).*

RESPONSE o

We have lengthened our discussion, referring to published work that proposes other functions for RPGR and addressing the limitations of our study.

Minor Comment p *References 18 and 19 are incorrectly cited and seem irrelevant to the statements made by the authors. Please revise.*

RESPONSE p

Reference 18 describes how the N-terminal half of Gelsolin is active, as demonstrated by actin severing (absence of fibroblast ruffling, loss of stress fibres and increased cell deformability). The reviewers are correct in that reference 18 does not demonstrate the process by which the N-terminal regions are exposed and activated. For this we also reference Burtnick et al (2004).

Reference 19 describes the original experiment that used triton to isolate F-actin from G-actin. This was used in the context of separating active and inactive gelsolin by Finkelstein et al (2010). We have amended this reference, and thank the reviewer for pointing this out.

We believe that we have addressed all of the referees' comments, and that the revisions now included substantially improve the paper. In particular, we believe the new data from the ColP experiments are in themselves significant contributions to the RPGR field. We hope therefore that you will be satisfied with the changes we have made to the manuscript.

Reviewers' Comments:

Reviewer #1:

None

Reviewer #3:

Remarks to the Author:

The authors have addressed the comments/suggestions made by this reviewer, and have much improved the quality of the manuscript.